# Forecasting Stochastic Volatility Characteristics for the Financial Fossil Oil Market Densities

Per Bjarte Solibakke 

Faculty of Economics and Management, Norwegian University of Science and Technology (NTNU), 6025 Aalesund, Norway; per.b.solibakke@ntnu.no; Tel.: +47-70161427

**Abstract:** This paper builds and implements multifactor stochastic volatility models for the international oil/energy markets (Brent oil and WTI oil) for the period 2011–2021. The main objective is to make step ahead volatility predictions for the front month contracts followed by an implication discussion for the market (differences) and observed data dependence important for market participants, implying predictability. The paper estimates multifactor stochastic volatility models for both contracts giving access to a long-simulated realization of the state vector with associated contract movements. The realization establishes a functional form of the conditional distributions, which are evaluated on observed data giving the conditional mean function for the volatility factors at the data points (nonlinear Kalman filter). For both Brent and WTI oil contracts, the first factor is a slow-moving persistent factor while the second factor is a fast-moving immediate mean reverting factor. The negative correlation between the mean and volatility suggests higher volatilities from negative price movements. The results indicate that holding volatility as an asset of its own is insurance against market crashes as well as being an excellent diversification instrument. Furthermore, the volatility data dependence is strong, indicating predictability. Hence, using the Kalman filter from a realization of an optimal multifactor SV model visualizes the latent step ahead volatility paths, and the data dependence gives access to accurate static forecasts. The results extend market transparency and make it easier to implement risk management including derivative trading (including swaps).

**Keywords:** energy; forecasting volatility; Markov Chain Monte Carlo (MCMC) simulations; projection-reprojection; stochastic volatility models

## 1. Introduction

This research develops and evaluates multifactor scientific stochastic volatility (SV) models for the purpose of predicting the volatility of the fossil oil market. At its most basics, volatility is a statistical measure of the dispersion of price movements for a given asset over a particular period. Typically, when prices fluctuate strongly and the associated bid–ask spreads are wide, the latent volatility is high. Market participants understand that the higher the volatility, the riskier the single contract. The adoption of any volatility model requires the ability to forecast future price movements. Internationally, a volatility model has been used to forecast the absolute magnitude of price fluctuations, quantiles, and full densities. The fact that asset volatility is not directly observable is one of its distinguishing characteristics (latent). Volatility's unpredictability makes evaluating the forecasting performance of volatility models difficult. However, when developing risk management strategies such as market selection, derivatives and hedging, market-making, and market timing, understanding the empirical features of future pricing is critical. Volatility predictability is critical for success in all these tasks. When it comes to other financial markets, portfolio studies have shown that as volatility rises, risk rises, and portfolio returns fall. If an asset manager adds more assets to his portfolio, the new assets diversify the portfolio if they do not covary (correlation less than 1) with the existing assets. As a result, portfolios imply diversification, emphasizing the need of asset allocation. The use

of derivatives promotes hedging, a risk-reduction approach that necessitates a thorough understanding of how to value derivatives and which risks should and should not be hedged. A risk manager will typically wish to know the contract volatility as maturity approaches for hedging purposes. In the Black–Scholes Model, the volatility measure is the only parameter that requires estimation. Estimating parameters (*u* and *d*) in a binomial model may also benefit from the volatility estimations. There are numerous reasons to trade volatility as a commodity. Higher (lower) volatility, in general, boosts (decreases) derivative prices. As a result, if the expected volatility is reducing, market players will sell (purchase) both call and put option contract holdings that are not part of speculative or hedging positions (increasing) (Alexander 2008). The fact that volatility and energy price changes are inversely associated suggests portfolio diversification as well as market collapse insurance. A market maker can also adjust his bid–ask spread if he believes future volatility will change. When volatility grows (falls), the bid–ask spread usually increases (decreases). The major stylized facts of asset, currency, and commodity price variations can be explained using SV models, which have a basic and intuitive framework. These models are not functions of purely observables enabling multiple shocks (simulations). The three main theoretical motivations for the of SV models: (1) random news item is constantly changing (unpredictable events), (2) time deformation (Clark 1973; Ané and Geman 2000), and (3) the approximation to a diffusion process for a continuous-time volatility variable (Hull and White 1987). Furthermore, the observed regularly shifting volatility is an additional motivation for SV models. Moreover, the models show well-known international volatility characteristics, for example, clustering and persistence (data dependence) which facilitate predictions. In financial markets, time-varying volatility is common, and market participants who understand the dynamic behavior of volatility are more likely to have accurate predictions about future prices and risks.

The SV implementation tries to describe how volatility evolves over time. Given that volatility is a non-traded instrument with inaccurate estimates, it can be viewed as a latent variable that can be modelled and predicted based on its direct impact on return magnitude. Risks can alter in complex ways over time; thus, it is only logical to create multifactor stochastic models to explain the temporal evolution of volatility. The method implements Chernozhukov and Hong's (Chernozhukov and Hong 2003) MCMC estimator, which is said to be significantly better than traditional derivative-based hill climbing optimizers for this stochastic class of problems. Furthermore, the normalized value of the objective function is asymptotically $\chi^2$ distributed given proper structural model specification (and the degrees of freedom is specified). The paper focuses on Gallant and Tauchen's (Gallant and Tauchen 1992, 1998, 2010a) Bayesian Markov Chain Monte Carlo (MCMC) modeling technique for implementing multivariate statistical models developed from scientific concerns. The method is a systematic approach to producing moment conditions for the generalized method of moments (GMM) estimator of structural model parameters (Gallant and Tauchen 2010b; Gallant and McCulloch 2011). Furthermore, the Chernozhukov and Hong (2003) estimator maintains model parameters in the region where predicted shares are positive for each observed price/expenditure vector. Furthermore, the methodology encourages limitations, inequity reductions, and the use of previous information that is informative (on the model parameters and functions). This article is organized as follows. Section 2 describes the methodology and explicitly describes the non-linear Kalman filter. Section 3 characterizes the Brent and WTI Oil front month contracts. Section 4 discusses the empirical major and minor findings, and Section 5 summarizes and concludes the paper.

## 2. Materials and Methods

The Stochastic Volatility (SV) Models[1] approach specifies the predictive distribution of price returns indirectly, rather than explicitly, through the structure of the model. The SV model has its own stochastic process, so the econometrician does not have to worry about the assumed one-step-ahead distribution of returns recorded over an arbitrary time interval. The starting point is Andersen et al. (2002)'s application, which considers the

well-known stochastic volatility diffusion for an observed stock price $S_t$ provided by $\frac{dS_t}{S_t} = (\mu + c(V_{1,t} + V_{2,t}))dt + \sqrt{V_{1,t}}dW_{1,t} + \sqrt{V_{2,t}}dW_{2,t}$, where the unobserved volatility processes $V_{i,t}$, $i = 1,2$, are either log linear or square root (affine). The $W_{1,t}$ and $W_{2,t}$ are standard Brownian motions that may be associated with corr$(dW_{1,t}, dW_{2,t}) = \rho$. Intuitively, the model has two shocks and one observable so that the current and past volatility are never observed precisely. With daily S&P500 stock index data from 1953 to 31 December 1996, Andersen et al. (2002) estimated both versions of the stochastic volatility model. Both versions of the SV model are outright rejected. Adding a jump component to a basic SV model, on the other hand, considerably improves the fit, reflecting two well-known features: fat non-Gaussian tails and persistent time-varying volatility. Chernov found promising results using an SV model with two stochastic volatility variables (Chernov et al. 2003). An affine configuration and a logarithmic setting are considered by the authors for the volatility index functions and factor dynamics. The models are based on daily Dow Jones Index data from 2 January 1953, through 16 July 1999. They discover that models with two volatility variables perform significantly better than models with simply one. They also discover that logarithmic two-volatility factor models beat affine jump diffusion models in terms of data fit. One volatility element is quite persistent, whereas the other is substantially mean reverting. The logarithmic model with two stochastic volatility factors is used in this research (Chernov et al. 2003). The Cholesky decomposition for consistency is used in the model to facilitate correlation between the factors. The key reason for correlation modeling is that it allows for asymmetry effects to be introduced (correlation between return and volatility innovations). Several alternative stochastic models are proposed in the international literature for oil contracts. However, SV models combined with a non-linear Kalman filter for functional forms and original data points are not available.[2]

The daily analyses cover the period from the end of 2011 until 5 February 2021, a total of 11 years giving 2443 daily price movements for the two front month future series. Price series are non-stationary and stationary logarithmic price changes from the two series are therefore used in the analysis. Any signs of successful SV-model implementations for the markets indicate random price change features and a minimum of weak-form market efficiency. Consequently, the markets are applicable for both enhanced risk management activities and derivative trading (including swaps). Summary statistics for the Brent and WTI oil contracts front month future markets are presented in Table 1 and Figure 1 reports their distributions and correlograms. Negative average price movements (negative drift) are reported for the raw data of both contracts. The standard deviation for the Brent Oil contracts (2.285) are lower than the WTI crude Oil contracts (2.909); therefore, reports lower the total risk. The maximum (19.1) and minimum (−28.0) numbers confirm lower risk for the Brent Oil contracts relative to the WTI crude oil contracts (22.4 and −56.9) contracts. Both front month oil contracts report a negative skewness coefficient, indicating that the return distributions have a negative skew. The kurtosis coefficients are relatively highly positive for both series, indicating a relatively peaked distributions with heavy tails. The WTI oil contract series has more peaks than the Brent oil contract, suggesting that the series has more observations close to the unconditional mean. The Cramer–von Mises normal test statistics and the quantile normal test suggest non-normal return distributions. Serial correlation in the mean equation is strong and the Ljung–Box $Q$-statistic (Ljung and Box 1978) is significant for both series. Volatility clustering using the Ljung–Box test statistic for squared returns ($Q^2$) and ARCH statistics seems to be present. The ADF (Dickey and Fuller 1979) and the Phillips–Perron (Phillips and Perron 1988) test statistics reject non-stationary series and the KPSS (Kwiatowski et al. 1992) statistic (12 lags) cannot reject stationary series.[3]

**Table 1.** Characteristics for Brent oil and WTI Crude oil Front Month Contracts 2011–2021.

| Brent Oil Front Month | | | | | | | | | |
|---|---|---|---|---|---|---|---|---|---|
| Mean (all)/ | Median | Max./ | Moment | Quantile | Quantile | Cramer- | Serial dependence | | VaR |
| M (-drop) | Std.dev. | Min. | Kurt/Skew | Kurt/Skew | Normal | von-Mises | Q(12) | Q²(12) | (1;2.5%) |
| −0.03419 | 0.06378 | 19.0774 | 23.2938 | 0.31680 | 12.4591 | 9.0716 | 17.9630 | 663.420 | −6.315% |
| −0.03828 | 2.28495 | −27.9761 | −1.15275 | −0.07439 | {0.0020} | {0.0000} | {0.1170} | {0.0000} | −9.588% |
| BDS-Z-statistic (ε = 1) | | | | | Phillips- | Augm. | ARCH | RESET | CVaR |
| m = 2 | m = 3 | m = 4 | m = 5 | KPSS | Perron | DF-test | (12) | (12;6) | (1;2.5%) |
| 13.8657 | 16.1690 | 18.3445 | 20.3371 | 0.07196 | −50.831 | −50.6842 | 372.232 | 104.961 | −9.588% |
| {0.0000} | {0.0000} | {0.0000} | {0.0000} | {0.2721} | {0.0000} | {0.0000} | {0.0000} | {0.0000} | −6.988% |
| **WTI Crude Oil Front Month** | | | | | | | | | |
| Mean (all)/ | Median | Max./ | Moment | Quantile | Quantile | Cramer- | Serial dependence | | VaR |
| M (-drop) | Std.dev. | Min. | Kurt/Skew | Kurt/Skew | Normal | von-Mises | Q(12) | Q²(12) | (1;2.5%) |
| −0.05629 | 0.03577 | 22.3940 | 77.63566 | 0.26708 | 8.0652 | 15.9531 | 110.240 | 1382.50 | −6.578% |
| −0.05688 | 2.90901 | −56.8589 | −3.38110 | −0.04463 | {0.0177} | {0.0000} | {0.1110} | {0.0000} | −13.616% |
| BDS-Z-statistic (ε = 1) | | | | | Phillips- | Augm. | ARCH | RESET | CVaR |
| m = 2 | m = 3 | m = 4 | m = 5 | KPSS | Perron | DF-test | (12) | (12;6) | (1;2.5%) |
| 14.8962 | 17.3605 | 19.5252 | 21.5017 | 0.04629 | −49.464 | −19.6282 | 514.024 | 158.376 | −13.616% |
| {0.0000} | {0.0000} | {0.0000} | {0.0000} | {0.2459} | {0.0000} | {0.0000} | {0.0000} | {0.0000} | −8.750% |

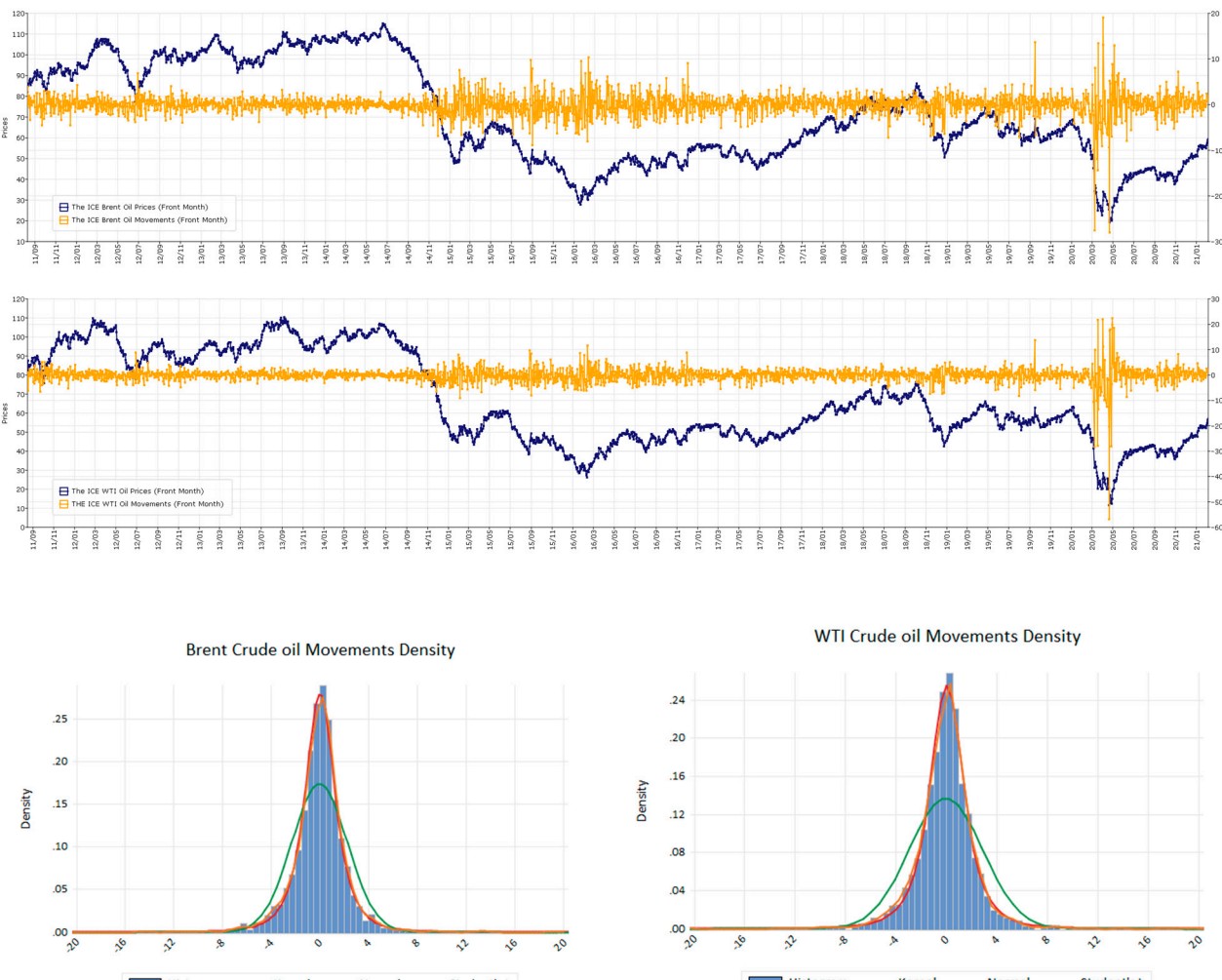

**Figure 1.** *Cont.*

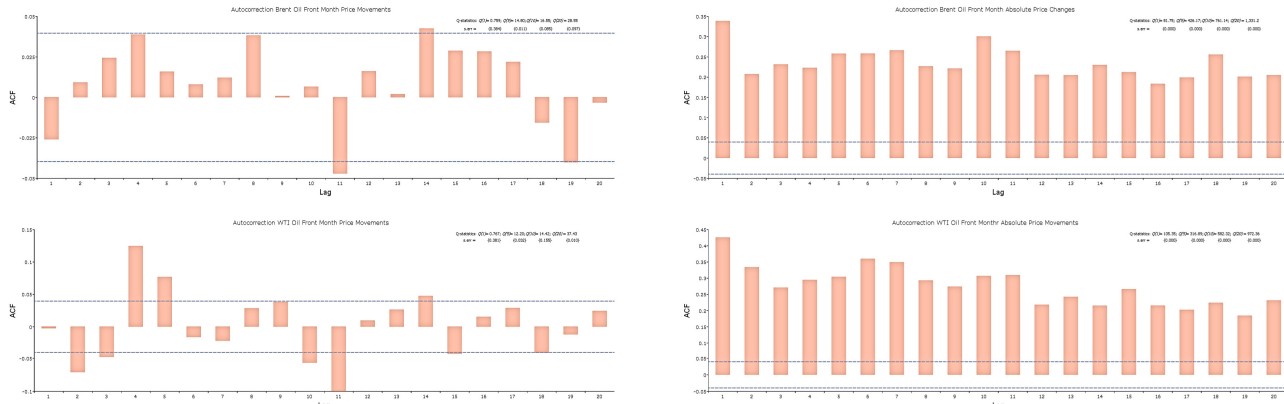

**Figure 1.** The ICE Brent Oil and WTI Oil Front Month contracts for the period 2011–2021.

The RESET (Ramsey 1969) test statistic, covering any departure from the assumptions of the maintained model, is significant (instability). Furthermore, the BDS independence test statistic (Brock et al. 1996) is a portmanteau test for time-based independence in a series. The probability of the distance between a pair of points being less or equal to epsilon ($\varepsilon$) should be constant (cm($\varepsilon$)). The BDS test statistics report highly significant data dependence for all integrals (m). Figure 1 (bottom) reports correlograms up to lag 20 for the daily price and squared/absolute price movements. The correlograms for daily price movements show only weak dependence while the correlogram for squared and absolute returns indicate substantial data dependence mainly in the form of serial correlation. The price change (log returns) data series show that the level of volatility seems to change randomly but shows a time varying nature typically for financial markets. Finally, the Quandt–Andrews breakpoint test (single breakpoint) (Andrews 1993) and the Bai (1997) and Bai and Perron (1998) multiple breakpoint test (sequential L + 1 breaks vs. L) report that both the single breakpoint test and the multiple breakpoint test cannot reject the null of zero breakpoints.[4] Furthermore, breakpoint unit root tests (trend and intercept) use an F-statistic augmented Dickey–Fuller equation (innovational outlier) test (Lumsdaine and Papell 1997; Lee and Strazicich 2003). Neither contract reports significant breakpoint statistics for intercepts, trends or break dummies. However, using the Schwarz (1978) criterion (BIC) with the innovational outlier specification and Dickey–Fuller (minimizing the *t*-statistic for $\alpha$ in the ADF-test) report one most likely COVID-19 break date for the Brent oil contracts (20200421) and for WTI crude oil contracts (20200427), suggesting that we cannot reject the hypothesis that the contracts have a unit root. These test statistics are also inherently related to whether the time series are affected by shocks. The COVID-19 period seems to signal temporary shocks for the two oil contracts. Moreover, to differentiate the price series twice will make the interpretation of the results much more complex. Therefore, the classical unit-root tests reported no breakpoint statistics and temporary shocks/COVID-19 signals; the manuscript assumes stationary return series.

The conditional moments are estimated using a statistical model for the $f(y|x)$ density, where y is the price movements and x represents lags of the series. Using their conditional moments, the SV model is estimated using the efficient method of moments (EMM) (Gallant and Tauchen 2010b; Gallant and McCulloch 2011). Table 2 reports the conditional moments from statistical moments (AR-GARCH) (panel A) and the stochastic SV model parameters (panel B). Columns for the mode, mean and standard deviation are reported. A test statistic ($\chi^2$) is reported at the bottom (panel B) for the SV model. The objective function accuracy is −5.1 and −3.6 for the Brent oil and WTI crude oil front month contracts, respectively, with associated $\chi^2$ test statistics of 0.121 (5 df) and 0.151 (5 df).[5] Figure 2 shows the MCMC log-posterior paths, which are an important measure for optimal solutions (success). The model passes the over-identified restrictions test at a 10% level, the chains are choppy, and the densities are close to normal (not reported), all of which indicate that the SV model is suitable for the fossil oil market series.

**Table 2.** Stochastic Volatility Model's optimal parameters.

**Panel A:**

Statistical Model SNP (11116000) BIC-fit; semi-parametric-GARCH model

| Var | SNP Coeff. | Mode and {Standard Error} The ICE Brent Oil | | The ICE WTI Oil | |
|---|---|---|---|---|---|
| *Hermite Polynoms* | | | | | |
| $\eta_1$ | a0[1] | −0.04087 | {0.0239} | −0.03243 | {0.0248} |
| $\eta_2$ | a0[2] | −0.15918 | {0.0196} | −0.17249 | {0.0255} |
| $\eta_3$ | a0[3] | −0.02248 | {0.0165} | −0.03431 | {0.0181} |
| $\eta_4$ | a0[4] | 0.10216 | {0.0162} | 0.07861 | {0.0182} |
| $\eta_5$ | a0[5] | 0.00464 | {0.0193} | 0.01815 | {0.0177} |
| $\eta_6$ | a0[6] | −0.07695 | {0.0139} | −0.07486 | {0.0125} |
| *Mean Equation (Correlation)* | | | | | |
| $\eta_7$ | b0[1] | 0.05108 | {0.0320} | 0.03047 | {0.0287} |
| $\eta_8$ | B(1,1) | −0.04853 | {0.0215} | −0.04274 | {0.0209} |
| *Variance Equation (Correlation)* | | | | | |
| $\eta_9$ | R0[1] | 0.07765 | {0.0159} | 0.09123 | {0.0133} |
| $\eta_{10}$ | P[1,1] | 0.18454 | {0.0524} | 0.13396 | {0.0728} |
| $\eta_{11}$ | Q[1,1] | −0.96608 | {0.0047} | 0.96131 | {0.0050} |
| $\eta_{12}$ | V[1,1] | −0.34431 | {0.0385} | −0.40901 | {0.0373} |
| $\eta_{13}$ | W[1,1] | 0.30542 | {0.0842} | 0.2378 | {0.1020} |
| *Model* | $s_n$ | 1.13023 | | 0.98116 | |
| *selection* | *aic* | 1.13556 | | 0.98648 | |
| *criterais:* | *bic* | 1.15100 | | 1.00193 | |
| *Largest eigenvalue for mean:* | | | 0.04853 | | 0.04274 |
| *Largest eigenvalue variance:* | | | 0.96737 | | 0.94205 |

**Panel B:**

Parameter values for Scientific Model.

| The ICE Brent Oil model | | | Standard | The ICE WTI Oil Model | | | Standard |
|---|---|---|---|---|---|---|---|
| $\theta$ | Mode | Mean | error | $\theta$ | Mode | Mean | error |
| *a0* | 0.048340 | 0.045396 | 0.008082 | *a0* | 0.047852 | 0.038677 | 0.018611 |
| *a1* | −0.057129 | −0.053005 | 0.003842 | *a1* | −0.063477 | −0.043269 | 0.022061 |
| *b0* | 0.141240 | 0.152300 | 0.007916 | *b0* | 0.447270 | 0.437200 | 0.037493 |
| *b1* | 0.830930 | 0.831160 | 0.001875 | *b1* | 0.985530 | 0.986230 | 0.002146 |
| *c1* | 0 | 0 | 0 | *c1* | 0 | 0 | 0 |
| *s1* | 0.165310 | 0.164670 | 0.000869 | *s1* | 0.052612 | 0.051506 | 0.003467 |
| *s2* | 0.162140 | 0.160670 | 0.000782 | *s2* | 0.228150 | 0.229830 | 0.003395 |
| *r1* | −0.313600 | −0.318690 | 0.006753 | *r1* | −0.636470 | −0.634260 | 0.009521 |
| *r2* | 0.078979 | 0.086698 | 0.008793 | *r2* | −0.120610 | −0.113660 | 0.006984 |
| Distributed Chi-square (freedoms) | | | $\chi^2(5)$ | Distributed Chi-square (freedoms) | | | $\chi^2(5)$ |
| Posterior | | | −5.0607 | Posterior at the mode | | | −3.5585 |
| Chi-square test | | | {0.1206} | Chi-square test statistic | | | {0.1507} |

The non-linear Kalman filter technique, moving backwards to infer the unobserved state vector from the observed process, is numerically intensive but straightforward (Hamilton 1994). From the long-simulated realization of the state vector, the AR-GARCH methodology obtains a convenient representation of one-step ahead conditional variance $\hat{\sigma}_t^2$ of $\hat{y}_{t+1}$ given $\{\hat{y}_\tau\}_{\tau=1}^t$. From simple regressions for $V_i$ $i = 1, 2$ on $\hat{\sigma}_t^2$, $\hat{y}_t$ and $|\hat{y}_\tau|$ and a generous number of lags of these series, the calibrated functions that give step ahead values of $V_{i,t}|\{y_\tau\}_{\tau=1}^t$, $t = 1, 2$ on the observed data series are constructed. The values for $V_i$, $i = 1,2$ are constructed at all observed data points.

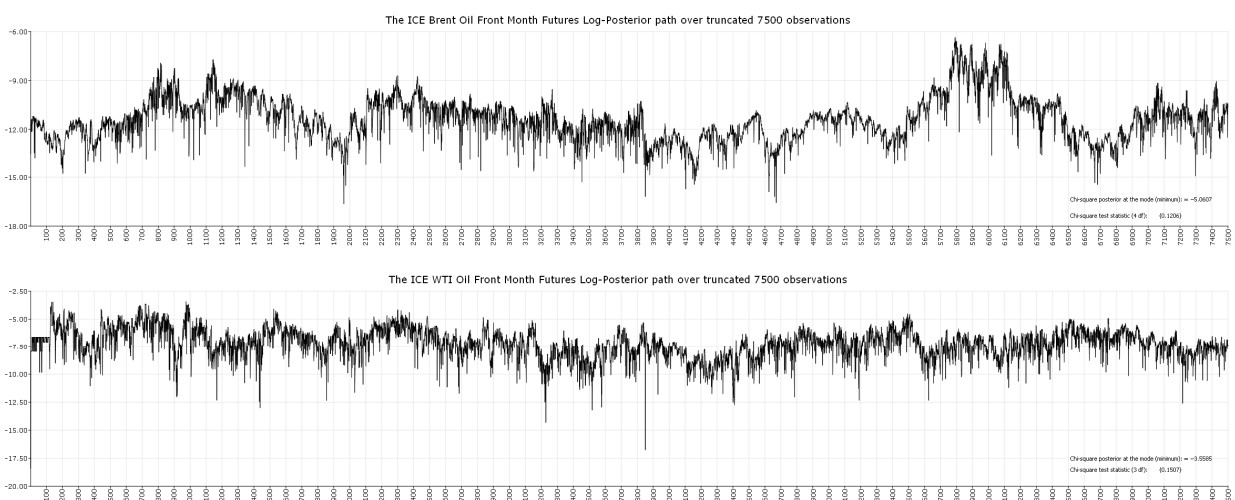

**Figure 2.** MCMC Posterior Chain from 250 *k* Optimal Stochastic Volatility Model (*R* = 75.000).

### 3. Results

#### 3.1. The Stochastic Volatility Factors

Table 2 reports the optimal parameters for the scientific models. The mean drift is similar (0.048), and the negative serial correlation is quite strong for WTI oil ($-0.063$ vs. $-0.057$). For the volatility, the parameter $b_0$ measures the constant volatility and $b_1$ reports the serial correlation. Brent oil (WTI crude oil) reports a relatively low (high) constant volatility of 0.14 (0.45), indicating that the constant information flow (or trading volume) is higher for WTI crude oil. The serial correlation for WTI oil (0.986) is clearly higher than Brent oil (0.831), indicating more information from the WTI volatility lags. The high correlation will probably also report higher data dependence. Furthermore, while Brent oil reports significant and similar parameters for volatility factor 1 ($V_1$) and factor 2 ($V_2$) of approximately 0.16, the WTI oil contracts report a factor of 0.05 for factor 1 and 0.228 for factor 2. The WTI oil volatility factor will therefore probably report a calm factor 1 ($V_1$) and a choppier factor 2 ($V_2$), giving larger volatility responses to price movement shocks. Finally, the correlation factor between the mean and volatility ($\rho_1$) is negative, indicating larger volatility for negative price movements. The WTI crude oil contracts report a correlation of $-0.636$, much higher than Brent oil contracts of $-0.314$, indicating that negative price movements increase WTI volatility more than Brent oil contract volatility.

For the way back to reality, the SV model simulation together with the non-linear Kalman filter approach establish the calibrated functions on all observed data points. The original data points, which now contain returns and volatility factors, are observable for all participants. For the period 2011–2021, Figure 3 reports factor 1 ($V_1$), factor 2 ($V_2$) and the $e^{(V_1+V_2)}$ for the observed data points for the ICE Brent Oil (top) and WTI crude oil (bottom), respectively. Interestingly, $V_1$ is a slow-moving, persistent volatility factor while $V_2$ is fast moving and the mean reverting factor. From the plots, the volatility $e^{(V_1+V_2)}$, seems more influenced by the $V_1$ factor than $V_2$. For visibility and a clearer and more interpretative evaluation of the factors, Figure 4 reports the last 60 days of these series in 2020/21 for the latent volatility factors for observed data points (21/01/2020–5/2/2021). It seems quite clear that the slowly persistent factor $V_1$, leads the reprojected yearly volatility for both series. For the period from March to May 2020 (COVID-19 outbreak), it is the slow-moving factor ($V_1$), showing higher persistence, that seems to show the main contribution to the yearly volatilities for both contracts. Additionally, for this period, the $V_2$ factor moves much faster, showing strong mean reversion, absorbing the shocks. The Brent oil contracts show both a higher and more choppy volatility. However, shocks seem to move the volatility higher for WTI oil contracts than Brent oil contracts. From the non-linear Kalman filter technique, Figures 3 and 4 also report the numerous and simulation-based ordinary least square number ($R^2$) for Brent oil and WTI crude oil contracts at a level of $V_i$, where i = 1, 2.

For $V_1$ ($V_2$) the R$^2$ is 71.5% (5.1%) and 98.3% (6.7%) for the Brent oil and the WTI crude oil contracts, respectively. Interestingly, the WTI crude oil contracts are both smoother and lower than the Brent oil contracts (also suggested by higher R$^2$).

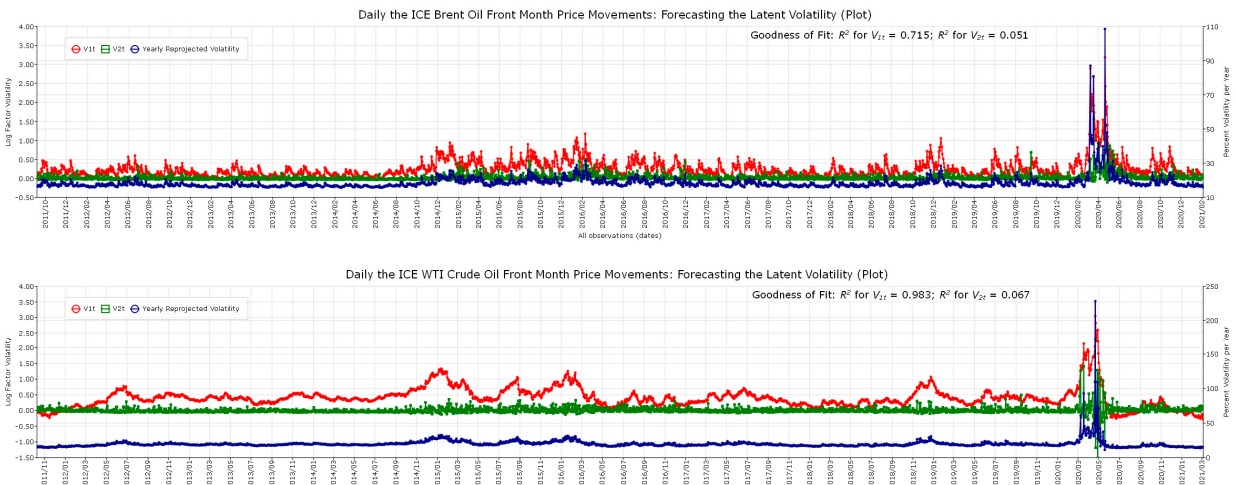

**Figure 3.** Daily Conditional Volatility from Observables and Kalman Filtered Volatility.

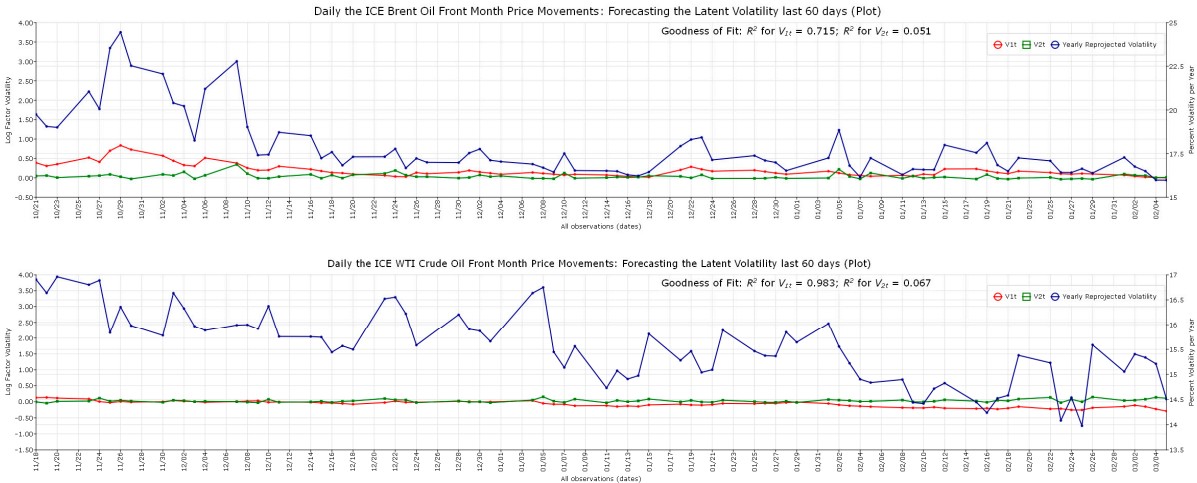

**Figure 4.** The ICE Front Month Brent Oil (**top**) and WTI Crude Oil (**bottom**) Factor Volatility Paths (last 60 days).

Table 3 reports characteristics for the nonlinear Kalman filtered volatility $\left( e^{(V_1+V_2)} \right)$. The statics report stationarity, non-normality with high kurtosis and positive skew, and some interesting volatility differences between Brent and WTI oil. The average yearly volatility is around 19% for both contracts, while the maximum daily volatility is much higher for WTI (228.2) than for Brent oil (108.5) contracts. The RESET (12,6) (Ramsey 1969) test (specification errors) suggests non-stable correlation parameters up to 12 lags. Table 3 and associated figures will be used for additional results in sub-sections below.

<div align="center">Table 3. Volatility Characteristics for the Oil Front Month Contracts.</div>

**Brent Oil Front Month**

| Mean (all)/ M (-drop) | Median Std.dev. | Maximum/ Minimum | Moment Kurt/Skew | Quantile Kurt/Skew | Quantile Normal | Cramer-von-Mises | Anderson-Darling | Serial dep. Q(12) |
|---|---|---|---|---|---|---|---|---|
| 18.4812 | 17.5436 | 108.53665 | 162.801 | 0.21031 | 40.2364 | 53.5048 | 296.842 | 8903.80 |
| | 4.0973 | 15.6741 | 10.0512 | 0.29755 | {0.0000} | {0.0000} | {0.0000} | {0.0000} |
| BDS-Z-statistic (ε = 1) | | | | | Phillips-Perron | Augm. DF-test | Breusch-Godfrey LM t. | |
| m = 2 | m = 3 | m = 4 | m = 5 | m = 6 | | | 10 lags | 20 lags |
| 44.3118 | 45.1030 | 44.7748 | 44.5608 | 44.7548 | −29.7032 | −4.552498 | 1513.60 | 1532.44 |
| {0.0000} | {0.0000} | {0.0000} | {0.0000} | {0.0000} | {0.0000} | {0.0000} | {0.0000} | {0.0000} |

**WTI Crude Oil Front Month**

| Mean (all)/ M (-drop) | Median Std.dev. | Maximum/ Minimum | Moment Kurt/Skew | Quantile Kurt/Skew | Quantile Normal | Cramer-von-Mises | Anderson-Darling | Serial dep. Q(12) |
|---|---|---|---|---|---|---|---|---|
| 19.9114 | 18.8897 | 228.20711 | 497.53830 | 0.42812 | 32.1454 | 57.8702 | 313.840 | 6198.00 |
| | 6.39430 | 11.80940 | 17.54151 | 0.18370 | {0.0000} | {0.0000} | {0.0000} | {0.0000} |
| BDS-Z-statistic (ε = 1) | | | | | Phillips-Perron | Augm. DF-test | Breusch-Godfrey LM t. | |
| m = 2 | m = 3 | m = 4 | m = 5 | m = 6 | | | 10 lags | 20 lags |
| 56.5300 | 56.1900 | 55.2791 | 54.8702 | 55.0307 | −47.2572 | −4.88025 | 1170.06 | 1210.83 |
| {0.0000} | {0.0000} | {0.0000} | {0.0000} | {0.0000} | {0.0000} | {0.0000} | {0.0000} | {0.0000} |

*3.2. Empirical Facts for the Volatility of Oil Contracts*

When comparing Figures 1 and 3, it seems clear that as returns become wider (narrower), volatility rises (decreases). Furthermore, turbulent (wide returns) days are more likely to be followed by other turbulent days, and calm (narrow returns) days are more likely to be followed by other calm days (Baillie et al. 1996). These properties will influence future volatility expectations. Volatility shows clustering/persistence if today's return has a large effect on the forecast variance for many periods in the future. From Table 3, the Q-statistic (Ljung and Box 1978) and the Breusch–Godfrey Lagrange multiplier test (Godfrey 1988) together with the correlograms in Figure 5(bottom) report expected dependencies up to 40 lags. Furthermore, the BDS independence test statistic (Brock et al. 1996), where $\varepsilon$ is one standard deviation and the number of dimensions (m) is six, reports that for both the Brent oil and WTI crude oil contracts, the volatility data strongly reject the hypothesis that the observations are independent. The WTI crude oil contracts show the highest BDS z-statistic dependence. An indication of serial correlation in volatility is also visible from the coefficient $b_1$ in Table 2. The table shows that the serial correlation is lower for the Brent oil ($b_1 = 0.828$) than for the WTI oil ($b_1 = 0.983$) contracts. The dependence on history therefore seems more profound for the WTI oil contracts. Results showing $b_1 > 0.8$ will indicate a form of volatility clustering. This is also visible in the above Figure 3 for WTI oil, which show longer periods of high/low volatility (less choppy).

Despite the fact that the volatility process has long dependencies, it seems to return to its mean. That is, the volatility process possesses mean reversion (stationarity). Table 3 reports the KPSS, Phillips–Perron, and augmented Dickey–Fuller test statistics. For both Brent and WTI oil front month contracts, all these test statistics report mean reversion. For Brent oil (WTI oil), the KPSS statistics are 0.108 {0.125} (0.109 {0.128}), the Phillips–Perron tests are −29.7 {0.0} (−47.3 {0.0}), and the ADF statistics are −4.6 {0.0} (−4.88 {0.0}), where t-statistics are reported in {}.

Furthermore, the volatility seems to increase more from negative price movements than from positive ones (Engle and Ng 1993). For the SV model in Table 2 (panel B), this asymmetry is modelled by negative correlation between returns and volatility ($r_1$). The negative asymmetry is significant for both the Brent and WTI oil contracts. This negative correlation suggests that holding volatility as an asset class of its own provides market participants with excellent diversification and by the same token, provide insurance against market crashes.

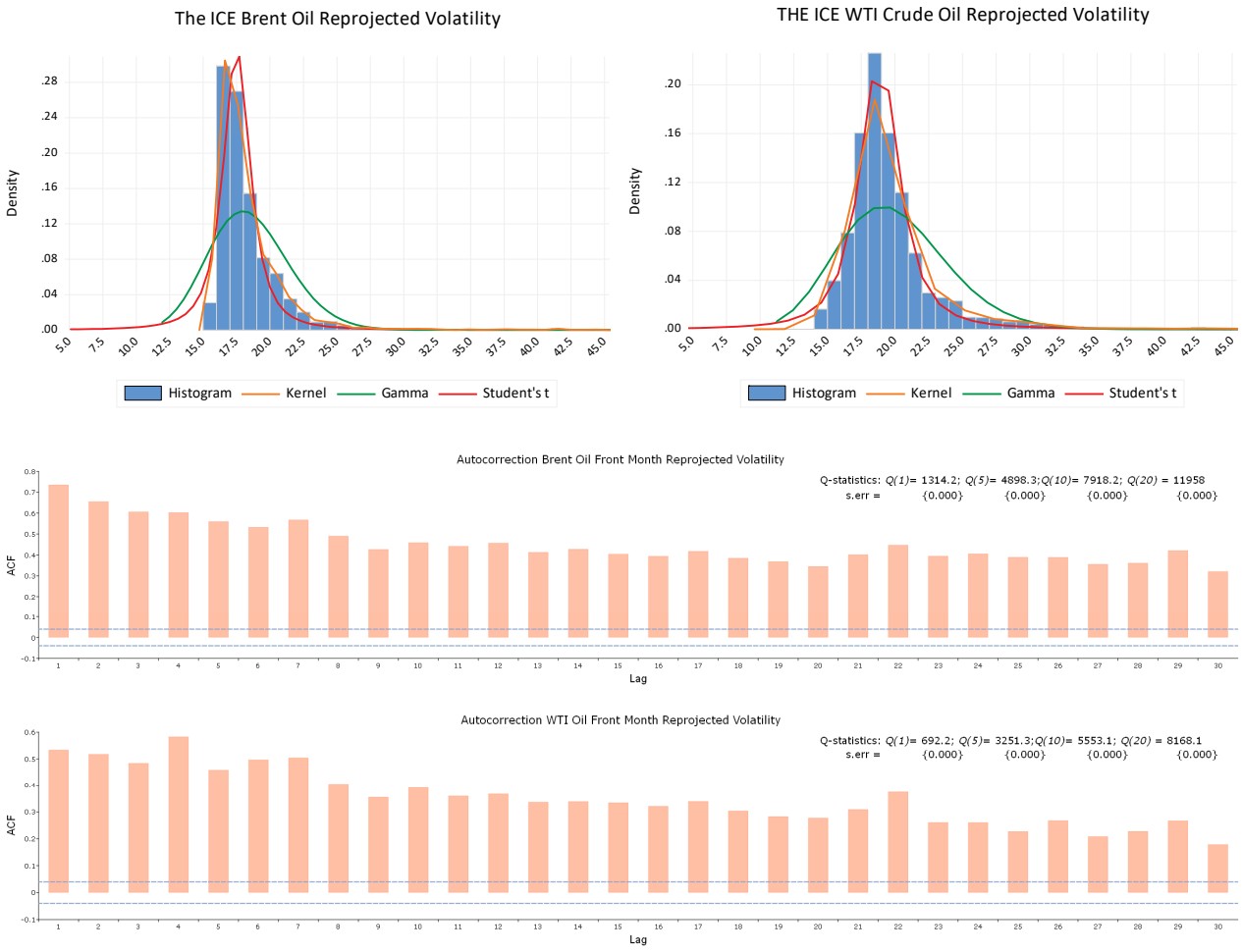

**Figure 5.** Daily Conditional Volatility from Observables and Kalman Filtered Volatility.

Volatility densities for both Brent oil and WTI oil contracts suggest lognormal densities. The density for the WTI oil versus the Brent oil volatility is lower and wider. However, the mean and standard deviation in Table 3 for WTI oil report a mean and standard deviation marginally higher than for Brent oil. These two results may suggest that the volatility for WTI oil is generally smoother and lower than for Brent Oil, but the shock responses seem higher (i.e., higher responses from the COVID-19 shocks are indicated by a higher maximum in Figure 3 (Table 3)). Furthermore, the Table 3 numbers show that the contracts report volatility that is non-normal, right-skewed, and indicate specification errors (RESET (Ramsey 1969). The distributions in Figure 5(top) strengthen these arguments, showing non-normal volatility distributions (log-normal).

### 3.3. Tail Probabilities

The power law, an alternative to assuming normal distributions, is applied to the nonlinear Kalman-filtered volatility $\left(e^{(V_1+V_2)}\right)$ for the Brent oil and WTI oil front month contracts. The power law asserts that given a large number of variables, the value of the variable, $v$, has the approximate property that when $x$ is large $Prob(v > x) = Kx^{-\alpha}$, where $K$ and $\alpha$ are constants. The relationship indicates that $\ln[Prob(v > x)] = \ln K - \alpha \ln x$, and charting $\ln[Prob(v > x)]$ against $\ln x$ is a test of whether it holds. The values for $\ln x$ and $\ln[Prob(v > x)]$ for the two energy contracts show that the logarithm of the probability of a change by more than x standard deviations is approximately linearly dependent in $\ln x$ for x ≥ 3. Hence, for both contracts, the power law holds for the volatility. Regressions show the estimates of $K$ and $\alpha$ are as follows: for Brent oil (WTI oil) contracts $K = e^{-1.9854}$ and $\alpha = -1.9735$ ($K = e^{-3.40169}$ and $\alpha = -1.43201$). A

probability estimate of a volatility greater than 3 [6] standard deviations is therefore $0.13732 \cdot 3^{-1.9735} = 0.0157(1.57\%)$ $\left[0.13732 \cdot 6^{-1.9735} = 0.0040(0.40\%)\right]$ $0.0333 \cdot 3^{-1.43201} = 0.0069(0.69\%)$ $\left[0.0333 \cdot 6^{-1.43201} = 0.0026(0.26\%)\right]$ for the Brent oil and the WTI oil contracts, respectively. The extreme value theory takes us another step (Gnedenko 1943)]. The $u$ is set to the 95 percentiles of the filtered volatility series of Brent oil ($u = 21.015$) and WTI crude oil ($u = 25.997$). The Brent oil contracts report optimal $\zeta = 2.0024$ and $\beta = 0.6486$ with an associated maximum value for the log-likelihood function of $-569.34$. The WTI oil contracts report optimal $\zeta = 3.934$ and $\beta = 0.5378$ with an associated maximum value for the log-likelihood function of $-354.71$. The probability that the Brent oil reprojected volatility will be greater than 25 (30) is 2.80% (1.22%). The VaR with 99% (99.9%) confidence limit is 31.70 (79.26). Hence, the 99.9% VaR estimate is about 0.730 times lower than the highest historic reprojected volatility (100.14). The 99% (99.9%) expected shortfall (ES) estimate is 57.13 (192.47). Furthermore, for the Brent oil contracts, the unconditional probability for a volatility greater than 21.015 ($u$) is equal to 0.112%. Similarly, the probability that the WTI oil contracts' reprojected volatility will be greater than 25 (30) is 6.62% (2.24%). The VaR with a 99% (99.9%) confidence limit is 36.13 (78.89). Hence, the 99.9% VaR estimate is about 0.3454 times higher than the highest historic filtered volatility for the WTI contracts (228.4). The 99% (99.9%) ES estimate is 56.44 (148.96). Finally, for the WTI crude oil contracts, the unconditional probability for volatility greater than 25.997 ($u$) is equal to 0.107%. As Var and ES are attempts to provide a single number that summarizes the volatility, tails give the market participants an indication of the risk to which they are exposed. The Brent oil contracts show that a daily volatility greater than 25 is only 2.80%. The WTI contracts show that a daily volatility greater than 25 is 6.62%. As a result, EVT and the power law, which reflect VaR and ES values, summarize tail features that imply market risks. Inverting the unconditional probability for volatility and putting a 1% limit for the change of the unconditional probability will list available investment options for market players.

### 3.4. Additonal Information and Contract Differences

From Table 2, both oil contracts report an appropriate posterior (bottom), and Figure 2 reports its path for Brent oil (top) and WTI oil (bottom). The paths are choppy and the MCMC chains seem appropriate (coefficient paths are not reported). The mean diffusion is quite similar, reporting positive drift and negative serial correlation. Both contracts therefore seem to show mean overreactions and reversion. The standard errors indicate a wider distribution for the two mean factors of WTI oil (varies more). Relative to the mean equation, the volatility model reports more differences between the contracts. The volatility constant and serial correlation is higher for WTI oil than Brent oil. However, while volatility factors 1 and 2 seem to have similar weights for Brent oil, factor 2 has considerably higher weight than factor 1 for the WTI oil. That is, volatility seems partly disconnected between the two oil contracts. The model also reports negative factor correlations between the mean and volatility (asymmetry), suggesting higher volatility from negative price shocks (highest for WTI oil). Correlation between the two volatility factors is insignificant. Finally, the Figure 5(top) (and Table 3) reports densities and characteristics for the nonlinear Kalman filtered volatility. The contracts report close means but the standard deviations together with maximum/minimum differs quite strongly. The WTI oil contract show a volatility range (maximum/minimum) much wider than for Brent oil. The finding is manifested in Figure 5; the density is lower and left and right tails are both bigger. The densities seem more log-normal than normal.

The volatility factors in Figures 3 and 4 seem to model two different flows of information to the market and the market participants. One slowly means reverting factor ($V_1$) provides volatility persistence and one ($V_2$) rapidly mean reverting factor provides for the tails (Gallant and Tauchen 2010b). Relative to brent oil, the WTI front month contract seems to have a marginal lower overall volatility. However, large volatility realization is found around the COVID-19 outbreak (from March to May 2020). For this research period, both

volatility factors, $V_1$ and $V_2$, show increases not seen either before or after COVID-19. The WTI oil reports the highest daily COVID-19 responses (see Figure 3, right axes).

The main differences between Brent oil and WTI oil volatility is therefore from the positive skewness and the maximum and minimum measures. Moreover, from Figure 5(top), the distribution for WTI oil is both wider and lower compared to Brent oil, indicating different volatility paths. Furthermore, Figure 5(bottom) reports long memory for the volatility series. The BDS data dependence measure in Table 3 reports higher dependence for WTI oil contracts at all lags (>55) than for Brent oil at all lags (>44). That is, the stronger data dependency for WTI oil contracts volatility should suggest improved predictability.

*3.5. Volatility Dependence and Predictability*

The SV model optimal parameter estimation gives several indications of predictability. The volatility parameter $b_1$ (see Table 2) is high, suggesting high correlation dependence on lagged volatility. The $b_1$ mode for the WTI oil contracts is 0.986 and Brent oil contracts is 0.831. Moreover, Table 3 (Q- and BDS statistic) and Figure 5(correlogram) above show considerable volatility data dependence for the Brent oil and WTI oil contracts. For example, the BDS statistic values in Table 3 is above 40 for Brent oil and above 50 for WTI oil contracts, and the correlogram with associated Q-statistics in Figure 5, reports high serial correlation up to lag 40. Still, the unit root test statistics in Table 3, KPSS, Phillips–Perron and ADF all suggest stationarity and mean reversion.

In case of a large market movement at any time before the risk horizon, the forecast needs to account for the SV model's stochastic processes, which are influenced by future random events. However, as already suggested, predicting volatility has profound influence on diversification and insurance against market crashes, making volatility an asset class valuable for market participants. Therefore, a static forecast for the Brent oil (top) and the WTI oil contracts (bottom) shown in Figure 6 and fit measures are reported in Table 4. The estimation period is from 2010 to 1 January 2020, and the forecasting period from 1 January 2020 to 5 February 2021. The RMSE and MAE are dependent on the scale of the dependent variable. However, the smaller the error, the better the forecasting ability. The MAPE and Theil measures are scale invariant. For a perfect fit, Theil's inequality coefficient is zero. For a "good" measure of fit, using the Theil inequality coefficient (bias, variance, and covariance portions), the bias and variance should be small, so that most of the bias has its focus on the covariance proportion. The covariance proportion for reprojected volatility for Brent oil (WTI oil) is 0.875 (0.821), indicating a reasonably good fit for a single asset. For the main contributor to reprojected volatility for both series, factor $V_1$, the covariance portion of the Theil inequality coefficient is even higher. For Brent oil (WTI oil), the $V_1$ factor shows a Theil's inequality coefficient close to zero (zero) and the covariance portion is as high as 0.933 (0.992).

Figure 6 shows static forecasts (one-step ahead) for the Brent oil (top) and the WTI crude oil (bottom) contracts for the Kalman filtered volatility as well as the two stochastic volatility sub-factors $V_1$ and $V_2$. For all factors, a step ahead distribution forecast can be constructed (not reported). Actuals and forecasts are reported as solid lines. Sup-norm bands are constructed by bootstrapping, using simulations to consider the sampling variation in the estimation of $\hat{f}(y|x)$. That is, changing the seed that generates densities and then to perform an impulse-response analysis to construct confidence intervals. For example, a 95% sup-norm confidence band is an $\varepsilon$-band around the mean profile that is just wide enough to contain 95% of the simulated profiles. The paper uses 1000 replications for the confidence intervals. Each plot reports Theil's measures at the top right corner. The Theil's covariance measures suggest static predictions that add new information to the market. For both Brent and WTI contracts, the Theil's covariance measure for the $V_1$ factor is high, suggesting derivative trading strategies as well as innovative products.

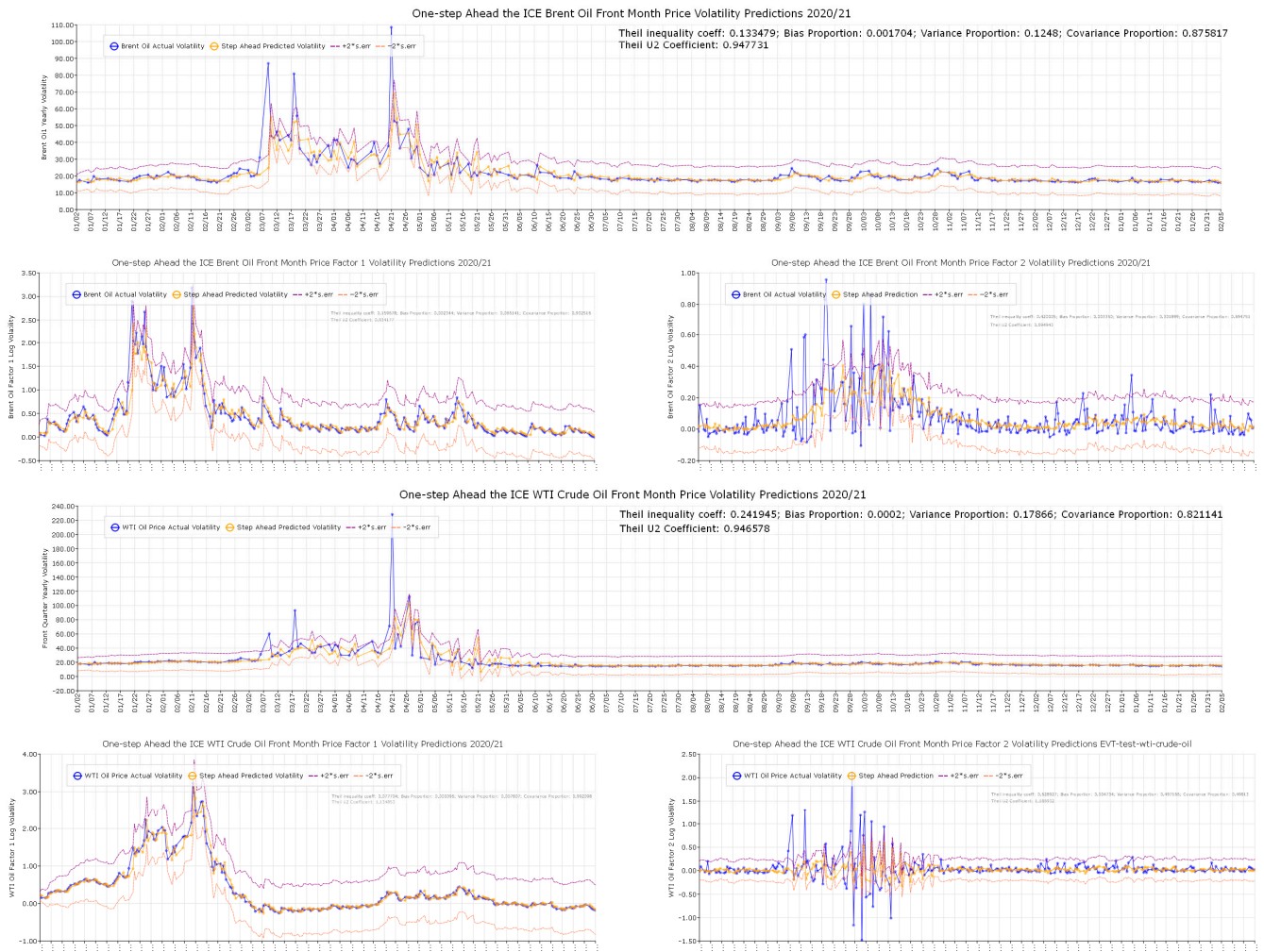

**Figure 6.** The ICE Brent Oil and WTI Crude Oil Front Month Contracts. Static Forecasts for 2020/21.

**Table 4.** Estimated stochastic volatility model and static forecasts fit measures.

| | | Factor 1 | | Factor 2 | | Reprojected | |
|---|---|---|---|---|---|---|---|
| **Contracts:** | **Error Measures:** | $V_{1t}$ | | $V_{2t}$ | | Volatility | |
| | Root Mean Square Error (RMSE) | 0.21111 | | 0.12444 | | 6.31367 | |
| | Mean absolute Error (MAE) | 0.11284 | | 0.07450 | | 2.31056 | |
| | Mean absolute percent error (MAPE) | 59.1436 | | 311.819 | | 7.77439 | |
| **Brent Oil** | Teil inequality coefficient (U1) | 0.15971 | | 0.42067 | | 0.13349 | |
| **Front Month** | Bias proportion | | 0.00238 | | 0.00342 | | 0.00171 |
| **Contracts:** | Variance Proportion | | 0.06532 | | 0.30304 | | 0.12260 |
| | Covariance Proportion | | 0.93229 | | 0.69354 | | 0.87569 |
| | Theil U2 Coefficient | 0.93418 | | 0.69494 | | 0.94773 | |
| | Symmetric MAPE | 30.70430 | | 120.238 | | 7.88895 | |
| | Root Mean Square Error (RMSE) | 0.11965 | | 0.24946 | | 12.6554 | |
| | Mean absolute Error (MAE) | 0.06640 | | 0.12347 | | 3.50976 | |
| | Mean absolute percent error (MAPE) | 100.822 | | 258.764 | | 11.4579 | |
| **WTI Crude Oil** | Teil inequality coefficient (U1) | 0.77040 | | 0.62893 | | 0.24195 | |
| **Front Month** | Bias proportion | | 0.00010 | | 0.00470 | | 0.00020 |
| **Contracts:** | Variance Proportion | | 0.00781 | | 0.49813 | | 0.17866 |
| | Covariance Proportion | | 0.99210 | | 0.49813 | | 0.82114 |
| | Theil U2 Coefficient | 1.13485 | | 1.18693 | | 0.94658 | |
| | Symmetric MAPE | 33.8832 | | 144.937 | | 10.7556 | |

## 4. Discussion

The major findings include firstly the appropriate SV model approach for modelling the ICE Oil front month contracts. The posterior at the mode with associated $\chi^2$ test statistics for Brent oil contracts (WTI oil contracts) is $-5.06$ {0.12} ($-3.55$ {0.15}), showing an appropriate modelling technique. That is, the combination of information determined by unpredictable events, time deformation (trading clock turns at different rates) and approximation to a diffusion process for continuous-time variables seem to be valid arguments for commodity futures markets.

Secondly, from the estimated SV model and the by-product, a long-simulated realization of the state vector, the non-linear Kalman filter gives a functional form of the conditional distribution. Reusing the AR-GARCH methodology and the calibrated functions form ordinary least squares (OLS), one-step ahead volatilities are calibrated for both Brent oil and WTI oil contracts. One-step ahead volatility is reported in Figure 3 and (whole period) and Figure 4 (last 60 days). Figures 3 and 4 show that Brent oil (WTI oil) has a choppier volatility. However, WTI oil is much more sensitive to shocks reporting a higher volatility for some periods (i.e., COVID-19). The contracts report a similar mean but different volatility factor paths. The volatility factors behave quite differently over time from approximately the same mean equation. That is, the two-factor stochastic volatility structure may add new information regarding objectives for both market crashes and diversification. For example. the WTI oil volatility is generally calmer than Brent oil. However, WTI oil is much more sensitive to shocks, showing an almost doubling of volatile relative to Brent oil during, for example, the COVID-19 outbreak.

Thirdly, the static forecasts evaluate and extend the SV models explicit factor volatilities, adding further insight into the predictability of volatility for commodities. Figure 6 reports Brent oil volatility forecasts (top) and WTI oil forecasts (bottom) for $V_1$, $V_2$ and $e^{(V_1+V_2)} \cdot \sqrt{252}$. The forecasts, actuals and a 95% confidence interval are reported. Table 4 reports fit statistics for $V_1$, $V_2$ and $e^{(V_1+V_2)} \cdot \sqrt{252}$. For single assets (not portfolios), the Theil's covariance fit for $e^{(V_1+V_2)} \cdot \sqrt{252}$ is acceptable, and for $V_1$, it is good (extremely good for WTI oil). For both contracts, the $V_2$ factor takes care of the tails, giving a lower covariance fit for the WTI oil $e^{(V_1+V_2)} \cdot \sqrt{252}$ volatility (as suggested earlier). A systematic approach to the all the factor volatilities may give valuable information for derivative trading (including swaps). Furthermore, the SV factor $V_1$ shows a higher predictability than the choppy factor $V_2$. Table 4 shows that $V_1$ has a Theil covariance portion of 93.3 and 99.2 for Brent oil and WTI oil, respectively. The $V_2$ factor shows a much lower covariance portion. Figures 3 and 4 show that the $V_1$ factor is clearly the major contributor to the yearly volatility, suggesting that a close focus on $V_1$ may improve derivative positioning.

Finally, the important negative correlation between the mean and volatility finding suggest higher volatility from negative price changes. This negativity indicates that holding volatility as an asset of its own has two major properties. First, it is insurance against market crashes, and secondly, it provides excellent diversification. This information may be even more valuable in combination with the volatility factor information. For example, trading forward volatility via calendar spreads provides a vega-hedge for forward start and cliquet options, and arbitrage traders (and hedge funds) can take positions on different volatilities on similar maturities.[6]

The minor findings are firstly the positive drift and negative serial correlation in the mean, suggesting daily overreaction followed by reversion next-trading day. Secondly, for the volatility, the SV model cannot show a significant correlation between volatility factors ($r_2$). Thirdly, for both oil contracts, their volatility shows that clustering, persistence, mean reversion, asymmetry, and long memory are all stylized facts. Furthermore, extreme tails using the power law and extreme value theory have been documented with probabilities and VaR/CVaR calculations for the series. Finally, the densities for both volatility series are more log-normal than normal.

## 5. Conclusions

The primary goal of this study was to define a volatility model that can foresee and capture commonly accepted stylized features about financial market volatility. Heavy tails, persistence, mean reversion, asymmetry (negative return innovations predict increased volatility), and long memory are among the stylized facts. The features imply that the volatility is highly data-dependent, suggesting information from previous periods as well as a model mean that is linear and a volatility that is non-linear. The paper demonstrates that the volatility of the energy markets exhibits all these characteristics, and that this data dependence suggests that volatility forecasting can help with risk management, portfolio timing and selection, market making, volatility trading, and derivative pricing for speculation and hedging.

For the Brent Oil and WTI crude oil contracts, the SV models show a positive mean drift and a negative mean serial correlation. For the mean and the price movements the result suggests negative dependence, meaning overreactions and reversions. The two oil contracts show the same properties. However, the standard error for WTI oil price movements suggests a lower and wider density. Moreover, several break point test statistics are used for both contracts, mostly signaling no break points. The Schwarz (1978) criterion reports shocks at *20200421* (*20200427*) for Brent oil (WTI crude oil) contracts. However, due to both classical unit-root tests and other breakpoint test statistics reporting stationary series, the analysis continues to use returns, which are important for the interpretation of manuscript results.

The volatility modelling for fossil oil contracts reports that a two-volatility factor model seems successful. One factor is slow moving and persistent ($V_1$), while the second factor ($V_2$) is fast moving and strongly mean reverting. However, the oil contracts report different volatility paths clearly visible in the SV model parameters. The constant ($b_0$) and the serial correlation ($b_1$) for the SV model volatility is higher for the WTI contracts than the Brent oil contracts. The result suggests systematic factor differences. The high $b_1$ coefficient for WTI oil (0.986) suggest better predictability. Furthermore, the high $b_0$ factor for the WTI oil volatility suggests a calmer WTI oil volatility. However, for the period from February to June 2020, the WTI oil contracts produce larger volatility spikes from the COVID-19 pandemic shock movements than the Brent oil contracts. Both contracts report stationarity and mean reversion for the volatility. The SV models therefore suggest almost no mean differences, but their latent volatility is systematically different. Furthermore, the $Q$ statistic and the BDS test statistic report heavy data dependence for both series (WTI oil highest) and the observed asymmetry suggests that volatility seems to grow higher from negative price movements than positive. The results suggest that to hold volatility as an asset class of its own provides market participants with insurance against oil market crashes as well as excellent diversification. Finally, the observed data dependence and volatility trading make volatility prediction an interesting exercise. Although price processes in energy markets are hardly predictable, the observed data dependence ($Q$ and BDS test statistics) suggest that the volatility can be estimated by means of observed past variations. A static prediction of the estimated volatility for the ICE front month Brent and WTI oil market contracts show a Theil's inequality coefficient close to zero and covariance portion of 87.5% and 82.1%, respectively, compared to actual volatility. The measures of fit may be further improved by using continuous prediction updates (i.e., daily). Furthermore, the main factor for the volatility, $V_1$ for the Brent oil (WTI oil) contracts, show a Theil inequality coefficient close to zero and an impressive covariance portion of 93.3% (99.2%). For market participants, a continuous SV model with associated volatility trading strategies may be applied to obtain superior positioning for the energy markets.

**Funding:** This research received no external funding.

**Institutional Review Board Statement:** Not applicable.

**Data Availability Statement:** The following are available online at the ICE.com 05022021 (registration required).

**Conflicts of Interest:** The author declares no conflict of interest.

## Notes

1     See Solibakke (2015, 2020) for an overview of the Brent oil front month contracts and a detailed definition and specification of a two-factor stochastic volatility model.

2     See also Oyuna and Yaobin (2021), Kyriakou et al. (2016), and Vo (2011) and references therein for a comparason between a classical Heston SV model and a GARCH-type volatility model, SV models with jumps and asian option, and SV models to extreact information interwined in stock and oil futures markets for risk prediction, respectively. However, all three papers build and report the SV models quite different from this paper's multifactor model.

3     Brent oil and WTI front month contract series is non-stationary reporting KPSS of 0.563 {0.0} and 0.604 {0.0}, ADF of $-1.845$ {0.68} and $-1.889$ {0.0}, and Phillips Perron $-1.966$ {0.619} and $-1.957$ {0.623}, respectively. In contrast the log price changes are stationary reporting KPSS of 0.072 {0.18} and 0.046 {0.25}, ADF of $-50.68$ {0.0} and $-19.63$ {0.0}, and Phillips Perron $-50.83$ {0.0} and $-49.46$ {0.0}, respectively.

4     Quandt-Andrews breakpoint test statistic report a value of 1.274 with an associated $p$-value of 0.957, suggesting no breakpoints. The multiple break test report an 0 vs. 1 $F$-statistic of 1.3498 with a Bai and Perron critical value of 8.58.

5     Estimation software is available from: https://www.aronaldg.org/webfiles (accessed on 10 August 2021).

6     See Alexander (2008), Market Risk Analsis, book part III.4.7 Trading Volatility, pp. 303–20.

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
