# Peer review of "Forecasting Stochastic Volatility Characteristics for the Financial Fossil Oil Market Densities"

_jrfm, doi:10.3390/jrfm14110510_

Round 1

Reviewer 1 Report

General comments:

  1. The approach to the investigated issues leads to rather mixed conclusions and although initially there is a good platform for investigation, the results shown are relatively vague and do not resolve the main question asked: how the volatility of oil price financial instruments behaves.
  2. The numbering of sections is chaotic and not properly organized. Please organize sections in a traditional logical manner, i.e. Introduction, Methodology, Data, Results, Discussion, and Conclusion.
  3. Given the rather complex issue investigated and the large number of parameters used to qualify the time series analyzed, we miss a more critical and scientific position about the methodology used, i.e. mainly SV and Kalman filter approach. This is especially the case regarding the absence of relevant literature on oil prices related investigations.  Section 2. Methods is a very clear example of this, please include more references to support the use of SV and Kalman filter methodologies on oil price-related research (please do also correct the numbering of another Section 2 following Data and SV model estimation).
  4. Although the methodology used to compare Brent oil and WTI instruments characteristics and prediction potential is driven through sound numerical tools, the reader misses a clear and concise description of the main volatility elements found for the two commodities, probably similar in most of the cases. The current drafting of the text regarding the properties of the two commodities is confusing and does not provide valuable insight on the main topic analyzed regarding oil price properties.
  5. Section 4. Market discussions.... is not located in the right place. The first part on volatility characteristics should be included in Section 3 Data. Please simplify figures and tables to show only relevant information for the investigation.
  6. It needs to be especially considered the conflicting evaluation of parameters due to incomparable units or difficulties in comparing similar attributes of the time series. This is clearly a main issue for improvement throughout the research paper.
  7. We miss in general a lighter document, better organized and with a clear section on Results (mainly on predicting ability of the SV analysis), another on Discussion, and a final section on Conclusions including a short description of how the significance level of time series prediction has been judged.

Particular comments

Section 1. Introduction.

  1. Line 29. The definition of volatility is not totally correct. Since it is a measure of dispersion, it describes the size of the distribution of values expected for a particular variable. Hence, in finance and investing, dispersion usually refers to the range of possible returns on investment. Please delete the statement ‘……price fluctuations are tightly bunched together…….’
  2. There is a lot of quotes throughout this section without any reference to existing literature. Please include valid references or delete. Examples of this are:
    • Lines 39-40 ‘When it comes……returns fall’.
    • Lines 50-51. ‘As a result, if expected volatility is reducing… hedging positions (increasing)’.
  3. Line 63. Please delete the statement made ‘Given that volatility is a non-traded instrument with inaccurate estimates’. This is incorrect since volatility is widely traded. The following are examples of related literature:
    1. Dutta, A. Oil and energy sector stock markets: An analysis of implied volatility indexes. J. Multinatl. Financ.Manag. 2018, 44, 61–68
    2. Aboura, S.; Chevallier, J. Leverage vs. feedback: Which Effect drives the oil market? Financ. Res. Lett. 2013,10, 131–141.
    3. Chen, Y.; Zou, Y. Examination on the relationship between OVX and crude oil price with Kalman filter. Procedia Comput. Sci. 2015, 55, 1359–1365.]

Section 2 (?). Data

13. Lines 123-126. As pointed out before, the comparison between Brent oil and WTI characteristics needs to be sharper and better organized. These lines are especially confusing and tedious and do not add value to the investigation. Please delete and rearrange the basic characteristics of the two oil products.

14. Lines 135-137. Results from ADF tests and KPSS are contradictory with most of the existing literature on unit root properties of oil prices. Most of the results of those tests indicate that ADF tests do not reject the null hypothesis of a unit root for oil and oil-related commodities. Similarly, the KPSS test, in which the null hypothesis is stationarity, indicates that the null hypothesis is commonly rejected. Please explain the results found and which could be the reason.

15. There is no reference to the existing literature on multiple structural breaks qualification and detection. This is not acceptable since the scope of research in structural breaks is vast and is now included as a main reference to assess the stationarity of time series. It is also inherently related to whether time series are affected by transitory or permanent shocks. Some examples of pace-setting research in this field are:

    1. Lumsdaine, R. and Papell, D., 1997. Multiple trend breaks and the unit root hypothesis, Review of Economics and Statistics, 79, 212-18.:
    2. Lee, J. and Strazicich, M.C., 2003. Minimum LM Unit Root Test with Two Structural Breaks. Review of Economics and Statistics, 63, pp. 1082-1089.
  1.  

Section 4.2 Volatility predictions.

16. In general, this should be the most important section leading to conclusions. Please reorganize it in a better way, simplify and make clear what is the added value of the research in terms of forecasting potential.

17. Lines 317-319. Please support with evidence or delete.

Section 5. Conclusion.

18. Interpretations of the results are provided, but a more critical position is required, e.g. linear and nonlinear relationships of oil prices are not mentioned. As it is known, the nature of the nonlinear relationships between crude oil prices and other variables has led to increased interest in multiscale modeling approaches such as mode decomposition techniques (EMD). A more profound exam of concluding elements is needed.

Author Response

The paper has gone through considerable changes after the first review. The paper sections now follow the MDPI standard with associated move of content enhancing readability. The following changes have been made relevant for reviewer 1:

  • The approach to the investigated issues leads to rather mixed conclusions and although initially there is a good platform for investigation, the results shown are relatively vague and do not resolve the main question asked: how the volatility of oil price financial instruments behaves.

Author:

The paper’s main objective is prediction of volatility of the oil contracts front month series. The platform is a SV model with explicit modelling of both the mean and volatility assuming stochastic factors. The estimated SV coefficients are from the SNP model moments using i.e., serial correlation in the mean and volatility and hermite functions for deviations from the normal distributions. The non-linear Kalman filter takes us back to the original data points using estimated regression functions from the long-simulated SV-model series. The behaviour of the volatility for the two oil contracts are shown in Figure 3 and 4. Note that the main focus of this paper is not on the behaviour of the volatility, which is well-known in the international literature, it is the step ahead predictability of volatility.

  • The numbering of sections is chaotic and not properly organized. Please organize sections in a traditional logical manner, i.e., Introduction, Methodology, Data, Results, Discussion, and Conclusion.

Author:

The paper is now rearranged into standard MPDI organization. Content is moved to other sections to follow the standard MDPI organization.

  • Given the rather complex issue investigated and the large number of parameters used to qualify the time series analysed, we miss a more critical and scientific position about the methodology used, i.e. mainly SV and Kalman filter approach. This is especially the case regarding the absence of relevant literature on oil prices related investigations.  Section 2. Methods is a very clear example of this, please include more references to support the use of SV and Kalman filter methodologies on oil price-related research (please do also correct the numbering of another Section 2 following Data and SV model estimation).

Author:

The SV and non-linear Kalman filter approaches are now well explained in section 2. Moreover, the section also references related SV/GARCH modelling of the oil contracts. References are from 2011 to 2021 with references therein. However, the combination of using MCMC SV modelling and non-linear Kalman filter for the movement back to original data points, have not been used for oil contracts. Solibakke (2020) has used this methodology for the equity market.

The standard MPDI organization together with re-organization/-writing hopefully improve both the international literature and use of the combination of MCMC SV models and the non-linear Kalman filters.

  • Although the methodology used to compare Brent oil and WTI instruments characteristics and prediction potential is driven through sound numerical tools, the reader misses a clear and concise description of the main volatility elements found for the two commodities, probably similar in most of the cases. The current drafting of the text regarding the properties of the two commodities is confusing and does not provide valuable insight on the main topic analysed regarding oil price properties.

Author:

 Using the standard MDPI organization as well as rewriting paragraphs focus on the volatility of oil contracts. However, in section 3 Results, the paper shows a similar mean (positive drift and negative serial correlation) but a clearly different volatility. This information is important for derivative trading in the oil business.

  • Section 4. Market discussions.... is not located in the right place. The first part on volatility characteristics should be included in Section 3 Data. Please simplify figures and tables to show only relevant information for the investigation.

Author:

Corrected, using standard MDPI organization of the paper.

  • It needs to be especially considered the conflicting evaluation of parameters due to incomparable units or difficulties in comparing similar attributes of the time series. This is clearly a main issue for improvement throughout the research paper.

Author:

Corrected, using standard MDPI organization of the paper. The evaluation of parameters is comparable, same length in time/observations (2.443), same market (the ICE London) and therefore synchronous, and both estimated using MCMC SV models and the non-linear Kalman filter.

  • We miss in general a lighter document, better organized and with a clear section on Results (mainly on predicting ability of the SV analysis), another on Discussion, and a final section on Conclusions including a short description of how the significance level of time series prediction has been judged.

Author:

Corrected, using standard MDPI organization of the paper.

Particular comments

Section 1. Introduction.

  • Line 29. The definition of volatility is not totally correct. Since it is a measure of dispersion, it describes the size of the distribution of values expected for a particular variable. Hence, in finance and investing, dispersion usually refers to the range of possible returns on investment. Please delete the statement ‘……price fluctuations are tightly bunched together…….’

Author:

 Corrected. The definition is reformulated and the statement ‘……price fluctuations are tightly bunched together…….’, is removed/changed.

  • There is a lot of quotes throughout this section without any reference to existing literature. Please include valid references or delete. Examples of this are:
    • Lines 39-40 ‘When it comes……returns fall’.
    • Lines 50-51. ‘As a result, if expected volatility is reducing… hedging positions (increasing)’.

Author:

 Corrected. References are added (Alexander, 2008) and changes are made to the abstract (pp. 39-40).

  • Line 63. Please delete the statement made ‘Given that volatility is a non-traded instrument with inaccurate estimates’. This is incorrect since volatility is widely traded. The following are examples of related literature:
    1. Dutta, A. Oil and energy sector stock markets: An analysis of implied volatility indexes. J. Multinatl. Financ.Manag. 2018, 44, 61–68
    2. Aboura, S.; Chevallier, J. Leverage vs. feedback: Which Effect drives the oil market? Financ. Res. Lett. 2013,10, 131–141.
    3. Chen, Y.; Zou, Y. Examination on the relationship between OVX and crude oil price with Kalman filter. Procedia Comput. Sci. 2015, 55, 1359–1365.]

Author:

The statement “…non-traded instrument with inaccurate estimates,…. is related to observed volatility time series. The existing time series are extruded form derivative prices (i.e. implied volatilities). The SV model does not use derivatives. The observations are traded market prices and volatility is estimated using stochastic factors and moments from the statistical SNP model. The estimates are therefore described as inaccurate.

The term is therefore from the applied methodology using underlying prices and not from derivatives (i.e., implied volatility). The sentence is therefore not deleted.

Section 2 (?). Data

  • Lines 123-126. As pointed out before, the comparison between Brent oil and WTI characteristics needs to be sharper and better organized. These lines are especially confusing and tedious and do not add value to the investigation. Please delete and rearrange the basic characteristics of the two oil products.

Author:

The lines are rearranged. The focus on Table 1 is important (negative drift, skewness, kurtosis, and serial correlation) because the SV model report positive drift but negative correlation.

  • Lines 135-137. Results from ADF tests and KPSS are contradictory with most of the existing literature on unit root properties of oil prices. Most of the results of those tests indicate that ADF tests do not reject the null hypothesis of a unit root for oil and oil-related commodities. Similarly, the KPSS test, in which the null hypothesis is stationarity, indicates that the null hypothesis is commonly rejected. Please explain the results found and which could be the reason.

Author:

The following results are available for prices and log price changes.

Prices:                                      Brent Oil                                 WTI oil

KPSS:                                     0.563668 {0.0000}                 0.604057 {0.0000}

ADF:                                       -1.845096 {0.6824}                -1.888575 {0.6601}

Phillips-Peron                          -1.966407 {0.6188}                -1.957299 {0.6237}

  • Non-stationary series for both contracts

Log Price changes:                  Brent Oil                                 WTI oil

KPSS:                                     0.071956 {0.1838}                 0.046287 {0.2459}

ADF:                                       -50.68422 {0.0000}                -19.62824 {0.0000}

Phillips-Peron                          -50.83143 {0.0000}                -49.46411 {0.0000}

  • Stationary series for both contracts

  • There is no reference to the existing literature on multiple structural breaks qualification and detection. This is not acceptable since the scope of research in structural breaks is vast and is now included as a main reference to assess the stationarity of time series. It is also inherently related to whether time series are affected by transitory or permanent shocks. Some examples of pace-setting research in this field are:
    1. Lumsdaine, R. and Papell, D., 1997. Multiple trend breaks and the unit root hypothesis, Review of Economics and Statistics, 79, 212-18.:
    2. Lee, J. and Strazicich, M.C., 2003. Minimum LM Unit Root Test with Two Structural Breaks. Review of Economics and Statistics, 63, pp. 1082-1089.

Author:

Both time series have been tested for 1) Quandt-Andrews Breakpoint test and 2) multiple breakpoint tests. The Q-A breakpoint test cannot reject no breakpoint within 15% trimmed data (Value: 1.274 {0.9570}). The multiple breakpoint test (Sequential L+1 breaks vs. L) using the methods outlined in Bai (1997) and Bai and Perron (1998). The 0 vs. 1 F-statistic and the Bai-Perron critical value is 1.349815 and 8.58, respectively. Hence, we cannot reject the null of 0 breakpoints.

Section 4.2 Volatility predictions.

  • In general, this should be the most important section leading to conclusions. Please reorganize it in a better way, simplify and make clear what is the added value of the research in terms of forecasting potential.

Author:

Reorganized using MDPI standard organization. The section is now in section 3.6 and is discussed in section 4.

  • Lines 317-319. Please support with evidence or delete.

Author:

Rewritten. However, basically, a stochastic process is random and should not be possible to predict (i.e. random walks).

Section 5. Conclusion.

  • Interpretations of the results are provided, but a more critical position is required, e.g., linear and nonlinear relationships of oil prices are not mentioned. As it is known, the nature of the nonlinear relationships between crude oil prices and other variables has led to increased interest in multiscale modelling approaches such as mode decomposition techniques (EMD). A more profound exam of concluding elements is needed.

Author:

 The focus of the paper is non-linear volatility prediction, not oil prices. The volatility is predicted from the non-linear Kalman filter’s functional move back to reality and data dependence using a static forecasting technique for step ahead volatility useful for derivative trading and swaps. The prediction is done for factor V1, factor V2 and exp(V1 + V2), possibly giving new information to the two oil front month markets.

Reviewer 2 Report

In this article, the author defines a volatility model that can predict and capture commonly accepted stylized features regarding the volatility of financial markets. The topic is really interesting and relevant to the journal. But the specific comments are as follows:

  • In the introduction, the purpose of the work should be more clearly indicated, as well as the rationale for the choice of the model that is used later in the manuscript;
  • The literature review should include articles from 2019-2021 (concerning, i.a., modeling volatility in the commodity market and methods of hedging against the risk of oil price fluctuations);
  • The designation of the sections is incorrect ("Data and SV model estimation" - Section 3; "Kalman filter approach" - Section 4; ....);
  • Figure 1 – the title in line 164 „WTUI Crude oil…” is incorrect;
  • Symbols of used variables should be corrected (lines 184-191);
  • None of the tables have been marked as "Table 4", but the author refers to them in the text (see line 323).

Author Response

The paper has gone through considerable changes after the first review. The paper sections now follow the MDPI standard with associated move of content enhancing readability. The following changes have been made relevant for reviewer 2:

Reviewer 2:

  • In the introduction, the purpose of the work should be more clearly indicated, as well as the rationale for the choice of the model that is used later in the manuscript.

Author:

The purpose is volatility prediction and the rationale for the choice of model is added to the introduction (Introduction: paragraph 1 page 2.).

  • The literature review should include articles from 2019-2021 (concerning, i.e., modelling volatility in the commodity market and methods of hedging against the risk of oil price fluctuations);

Author:

Stochastic volatility models for the oil market for 2019 - 2021 is well summarized in Oyuna and Yaobin, 2021 with refrenced therein. In addition, SV modelling is also well documented in Kyriakou et al., 2016 and Vo, 2011. All three papers are added to the text (Materials and Methods) and the reference list.

  • The designation of the sections is incorrect ("Data and SV model estimation" - Section 3; "Kalman filter approach" - Section 4; ....);

Author:

Corrected, following MDPI standard.

  • Figure 1 – the title in line 164 „WTUI Crude oil…” is incorrect;

Author:

Corrected.

  • Symbols of used variables should be corrected (lines 184-191);

Author:

Corrected. All symbols ++ are from MathType 7.4.4.516 (NTNU License)

  • None of the tables have been marked as "Table 4", but the author refers to them in the text (see line 323)

Author:

Table 4 should not exist. References should be to Table 3. Corrected in the text.

Round 2

Reviewer 1 Report

I am dissatisfied with the approach during the author's review. In general, the suggested corrections I made have not been taking into account at least in a careful and professional manner.

I send the updated comments to the previous author's response within the pdf document. Please take those into consideration this time otherwise I would understand there is no reason to continue reviewing this paper.

Author Response

Reviewer 1 writes: …, the suggested corrections I made have not been taking into account at least in a careful and professional manner.

Author: The author has made corrections to comments that is yellowed by the reviewer. There are room for interpretations, and I have tried to express my views as clear as possible. For example, for the break unit root tests, I have added an extra reference to the text and performed an actual Breakpoint Unit Root test for the contracts below.

Furthermore, I want to stress that this research is carefully and professionally done by Per B Solibakke. The paper idea is based on these research results and written by Per B Solibakke.

****

From the first review all new author comments are in background light grey.

  • The approach to the investigated issues leads to rather mixed conclusions and although initially there is a good platform for investigation, the results shown are relatively vague and do not resolve the main question asked: how the volatility of oil price financial instruments behaves.

Author:

The paper’s main objective is prediction of volatility of the oil contracts front month series. The platform is a SV model with explicit modelling of both the mean and volatility assuming stochastic factors. The estimated SV coefficients are from the SNP model moments using i.e., serial correlation in the mean and volatility and hermite functions for deviations from the normal distributions. The non-linear Kalman filter takes us back to the original data points using estimated regression functions from the long-simulated SV-model series. The behaviour of the volatility for the two oil contracts are shown in Figure 3 and 4. Note that the main focus of this paper is not on the behaviour of the volatility, which is well-known in the international literature, it is the step ahead predictability of volatility.

The author has added a section for the SV model parameters. Both parameter structure and difference between oil contracts are elaborated. The use of the non-linear Kalman filter approach for the move back to reality and the observed time series, is explained at the end of section 2 (methodology).

  • The numbering of sections is chaotic and not properly organized. Please organize sections in a traditional logical manner, i.e., Introduction, Methodology, Data, Results, Discussion, and Conclusion.

Author:

The paper is now rearranged into standard MPDI organization. Content is moved to other sections to follow the standard MDPI organization.

  • Given the rather complex issue investigated and the large number of parameters used to qualify the time series analysed, we miss a more critical and scientific position about the methodology used, i.e. mainly SV and Kalman filter approach. This is especially the case regarding the absence of relevant literature on oil prices related investigations.  Section 2. Methods is a very clear example of this, please include more references to support the use of SV and Kalman filter methodologies on oil price-related research (please do also correct the numbering of another Section 2 following Data and SV model estimation).

Author:

The SV and non-linear Kalman filter approaches are now well explained in section 2 and a section for the result of the SV model parameters are added in start of section 3. Moreover, the section also references related SV/GARCH modelling of the oil contracts. References are from 2011 to 2021 with references therein. However, the combination of using MCMC SV modelling and non-linear Kalman filter for the movement back to original data points, have not been used for oil contracts. Solibakke (2020) has used this methodology for the equity market.

The standard MPDI organization together with re-organization/-writing hopefully improve both the international literature and use of the combination of MCMC SV models and the non-linear Kalman filters.

  • Although the methodology used to compare Brent oil and WTI instruments characteristics and prediction potential is driven through sound numerical tools, the reader misses a clear and concise description of the main volatility elements found for the two commodities, probably similar in most of the cases. The current drafting of the text regarding the properties of the two commodities is confusing and does not provide valuable insight on the main topic analysed regarding oil price properties.

Author:

 Using the standard MDPI organization as well as rewriting paragraphs. The main focus of the current research is not the mean (price movements), but the volatility of the oil contracts.

The SV model and Figure 3 and 4 of the paper describe the volatility in detail from 2011- 2021 based on the parameters from the respective SV models.

For the mean:

However, in section 3 Results, the paper shows a similar mean (positive drift and negative serial correlation)

For the volatility:

The SV model parameters b0, b1, s1, s2 and r1 are significant and clearly different (Table 2 panel B). The result is different volatility paths.

The author has added an extra section in the begging of section 3 to complement the description of the SV model parameters.

  • Section 4. Market discussions.... is not located in the right place. The first part on volatility characteristics should be included in Section 3 Data. Please simplify figures and tables to show only relevant information for the investigation.

Author:

Corrected, using standard MDPI organization of the paper.

  • It needs to be especially considered the conflicting evaluation of parameters due to incomparable units or difficulties in comparing similar attributes of the time series. This is clearly a main issue for improvement throughout the research paper.

Author:

Corrected, using standard MDPI organization of the paper. The evaluation of parameters is comparable, same length in time/observations (2.443), same market (the ICE London) and therefore synchronous, and both estimated using MCMC SV models with the non-linear Kalman filter for functional values back to reality.

I do not understand the reviewer’s comment: “… conflicting evaluation of parameters due to incomparable units or difficulties in comparing similar attributes of the time series.”

If the reviewer yellowed the previous answer because I do not answer his comment, I do not understand what he is looking for or how I can meet his requirements!

  • We miss in general a lighter document, better organized and with a clear section on Results (mainly on predicting ability of the SV analysis), another on Discussion, and a final section on Conclusions including a short description of how the significance level of time series prediction has been judged.

Author:

Corrected, using standard MDPI organization of the paper.

Particular comments

Section 1. Introduction.

  • Line 29. The definition of volatility is not totally correct. Since it is a measure of dispersion, it describes the size of the distribution of values expected for a particular variable. Hence, in finance and investing, dispersion usually refers to the range of possible returns on investment. Please delete the statement ‘……price fluctuations are tightly bunched together…….’

Author:

 Corrected. The definition is reformulated and the statement ‘……price fluctuations are tightly bunched together…….’, is removed/changed with:

Typically, when prices fluctuate strongly, and the associated bid-ask spreads are wide, the latent volatility is high.

  • There is a lot of quotes throughout this section without any reference to existing literature. Please include valid references or delete. Examples of this are:
    • Lines 39-40 ‘When it comes……returns fall’.
    • Lines 50-51. ‘As a result, if expected volatility is reducing… hedging positions (increasing)’.

Author:

 Corrected. References are added (Alexander, 2008) and changes are made to the abstract (pp. 39-40).

  • Line 63. Please delete the statement made ‘Given that volatility is a non-traded instrument with inaccurate estimates’. This is incorrect since volatility is widely traded. The following are examples of related literature:
    1. Dutta, A. Oil and energy sector stock markets: An analysis of implied volatility indexes. J. Multinatl. Financ.Manag. 2018, 44, 61–68
    2. Aboura, S.; Chevallier, J. Leverage vs. feedback: Which Effect drives the oil market? Financ. Res. Lett. 2013,10, 131–141.
    3. Chen, Y.; Zou, Y. Examination on the relationship between OVX and crude oil price with Kalman filter. Procedia Comput. Sci. 2015, 55, 1359–1365.]

Author:

The statement “…non-traded instrument with inaccurate estimates,…. is related to non-observed volatility time series in financial markets. The only volatility time series reported are extracted from liquid option markets using the implied volatility from B&S formula. Therefore, existing time series are extruded form derivative prices. This research use an SV model using only price movements. Hence, the observations are traded market prices. The volatility is therefore estimated from stochastic factors and moments from statistical moments (AR_GARCH) models. The estimates are therefore described as inaccurate.

The term is therefore from the applied methodology using underlying prices and not from derivatives (i.e., implied volatility). The sentence is therefore clearly meaningful when only using asset prices and not derivatives.

 Section 2 (?). Data

  • Lines 123-126. As pointed out before, the comparison between Brent oil and WTI characteristics needs to be sharper and better organized. These lines are especially confusing and tedious and do not add value to the investigation. Please delete and rearrange the basic characteristics of the two oil products.

Author:

The lines are rearranged. The focus on Table 1 is important (negative drift, skewness, kurtosis, and serial correlation) because the SV model report positive drift and negative correlation. Relative to lines 123-126.

The front month products have negative average price movements (negative drift). The standard deviation for the Brent Oil contracts (2.285) are lower than the WTI crude Oil contracts (2.909), and therefore reports lower risk. The maximum (19.08) and minimum (-27.98) numbers confirm lower risk for the Brent Oil contracts relative to the WTI crude oil contracts (a maximum of 22.39 and a minimum of -56.86) contracts.

Rewritten. However, the numbers are important part of Table 1 and need to be mentioned in the text. The difference in for example max/min values for pric3e movements cannot be tedious? The difference is so large that it needs to be mentioned in the text.

  • Lines 135-137. Results from ADF tests and KPSS are contradictory with most of the existing literature on unit root properties of oil prices. Most of the results of those tests indicate that ADF tests do not reject the null hypothesis of a unit root for oil and oil-related commodities. Similarly, the KPSS test, in which the null hypothesis is stationarity, indicates that the null hypothesis is commonly rejected. Please explain the results found and which could be the reason.

Author:

The following results are available for prices and log price changes.

Prices:                                      Brent Oil                                 WTI oil

KPSS:                                     0.563668 {0.0000}                 0.604057 {0.0000}

ADF:                                       -1.845096 {0.6824}                -1.888575 {0.6601}

Phillips-Peron                          -1.966407 {0.6188}                -1.957299 {0.6237}

  • Non-stationary series for both contracts

Log Price changes:                  Brent Oil                                 WTI oil

KPSS:                                     0.071956 {0.1838}                 0.046287 {0.2459}

ADF:                                       -50.68422 {0.0000}                -19.62824 {0.0000}

Phillips-Peron                          -50.83143 {0.0000}                -49.46411 {0.0000}

  • Stationary series for both contracts

  • There is no reference to the existing literature on multiple structural breaks qualification and detection. This is not acceptable since the scope of research in structural breaks is vast and is now included as a main reference to assess the stationarity of time series. It is also inherently related to whether time series are affected by transitory or permanent shocks. Some examples of pace-setting research in this field are:
    1. Lumsdaine, R. and Papell, D., 1997. Multiple trend breaks and the unit root hypothesis, Review of Economics and Statistics, 79, 212-18.:
    2. Lee, J. and Strazicich, M.C., 2003. Minimum LM Unit Root Test with Two Structural Breaks. Review of Economics and Statistics, 63, pp. 1082-1089.

Author:

Both time series have been tested for 1) Quandt-Andrews Breakpoint test and 2) multiple breakpoint tests. The Q-A breakpoint test cannot reject no breakpoint within 15% trimmed data (Value: 1.274 {0.9570}). The multiple breakpoint test (Sequential L+1 breaks vs. L) using the methods outlined in Bai (1997) and Bai and Perron (1998). The 0 vs. 1 F-statistic and the Bai-Perron critical value is 1.349815 and 8.58, respectively. Hence, we cannot reject the null of 0 breakpoints.

We have also performed a Breakpoint unit root test for the price movements, using innovation outlier, with basic trend specification to Trend and Intercept, Breaking Trend Specification to Trend and Intercept, selecting Dickey Fuller min-t as the breakpoint selection and changing the Lag Length Method to F-statistic >(10, p-value 0.1). 

The results for Brent oil contracts are:

Null Hypothesis: SPOT_RETURNS has a unit root

Trend Specification: Trend and intercept

Break Specification: Trend and intercept

Break Type: Innovational outlier

Break Date: 8/26/2011

Break Selection: Minimize Dickey-Fuller t-statistic

Lag Length: 0 (Automatic - based on F-statistic selection, lagpval=0.1,

        maxlag=10)

t-Statistic

  Prob.*

Augmented Dickey-Fuller test statistic

-50.65383

< 0.01

Test critical values:

1% level

-5.719131

5% level

-5.175710

10% level

-4.893950

*Vogelsang (1993) asymptotic one-sided p-values.

Augmented Dickey-Fuller Test Equation

Dependent Variable: SPOT_RETURNS

Method: Least Squares

Date: 09/22/21   Time: 11:00

Sample (adjusted): 8/11/2011 2/05/2021

Included observations: 2440 after adjustments

Variable

Coefficient

Std. Error

t-Statistic

Prob.  

SPOT_RETURNS(-1)

-0.026065

0.020256

-1.286742

0.1983

C

0.507242

1.478336

0.343117

0.7315

TREND

-0.025903

0.217937

-0.118854

0.9054

INCPTBREAK

-0.266457

1.292484

-0.206159

0.8367

TRENDBREAK

0.025907

0.217937

0.118874

0.9054

BREAKDUM

0.721968

2.287294

0.315643

0.7523

R-squared

0.000844

    Mean dependent var

-0.035814

Adjusted R-squared

-0.001209

    S.D. dependent var

2.284011

S.E. of regression

2.285391

    Akaike info criterion

4.493408

Sum squared resid

12712.82

    Schwarz criterion

4.507669

Log likelihood

-5475.957

    Hannan-Quinn criter.

4.498592

F-statistic

0.410980

    Durbin-Watson stat

1.999912

Prob(F-statistic)

0.841442

  • No breakpoints are significant for Brent oil contracts.

The results for WTI oil contracts are:

Null Hypothesis: SPOT_RETURNS has a unit root

Trend Specification: Trend and intercept

Break Specification: Trend and intercept

Break Type: Innovational outlier

Break Date: 8/30/2011

Break Selection: Minimize Dickey-Fuller t-statistic

Lag Length: 0 (Automatic - based on F-statistic selection, lagpval=0.1,

        maxlag=10)

t-Statistic

  Prob.*

Augmented Dickey-Fuller test statistic

-49.45352

< 0.01

Test critical values:

1% level

-5.719131

5% level

-5.175710

10% level

-4.893950

*Vogelsang (1993) asymptotic one-sided p-values.

Augmented Dickey-Fuller Test Equation

Dependent Variable: SPOT_RETURNS

Method: Least Squares

Date: 09/22/21   Time: 11:17

Sample (adjusted): 8/17/2011 2/05/2021

Included observations: 2440 after adjustments

Variable

Coefficient

Std. Error

t-Statistic

Prob.  

SPOT_RETURNS(-1)

-0.002353

0.020269

-0.116088

0.9076

C

-1.695190

2.115555

-0.801298

0.4230

TREND

0.350919

0.375934

0.933457

0.3507

INCPTBREAK

-1.522190

1.793590

-0.848683

0.3961

TRENDBREAK

-0.350917

0.375934

-0.933452

0.3507

BREAKDUM

1.914828

2.914423

0.657018

0.5112

R-squared

0.000544

    Mean dependent var

-0.055739

Adjusted R-squared

-0.001509

    S.D. dependent var

2.909473

S.E. of regression

2.911667

    Akaike info criterion

4.977785

Sum squared resid

20634.98

    Schwarz criterion

4.992046

Log likelihood

-6066.897

    Hannan-Quinn criter.

4.982969

F-statistic

0.265115

    Durbin-Watson stat

2.000047

Prob(F-statistic)

0.932231

  • No breakpoints are significant for WTI oil contracts.

Banerjee, Anindya, Robin L. Lumsdaine, and James H. Stock (1992). “Recursive and Sequential Tests of the Unit-Root and

Trend-Break Hypotheses: Theory and International Evidence,” Journal of Business & Economic Statistics, 10,

  1. 271–287.

Section 4.2 Volatility predictions.

  • In general, this should be the most important section leading to conclusions. Please reorganize it in a better way, simplify and make clear what is the added value of the research in terms of forecasting potential.

Author:

Reorganized using MDPI standard organization. The section is now in section 3.6 and is discussed in section 4.

  • Lines 317-319. Please support with evidence or delete.

Author:

Rewritten. However, basically, a stochastic process is random and should not be possible to predict (i.e. random walks).

Section 5. Conclusion.

  • Interpretations of the results are provided, but a more critical position is required, e.g., linear and nonlinear relationships of oil prices are not mentioned. As it is known, the nature of the nonlinear relationships between crude oil prices and other variables has led to increased interest in multiscale modelling approaches such as mode decomposition techniques (EMD). A more profound exam of concluding elements is needed.

Author:

The focus of the paper is non-linear volatility prediction, not oil prices. The volatility is predicted from the non-linear Kalman filter’s functional move back to reality and data dependence using a static forecasting technique for step ahead volatility useful for derivative trading and swaps. The prediction is done for factor V1, factor V2 and exp(V1 + V2), possibly giving new information to the two oil front month markets.

The conclusion is rewritten trying to focus more on the relationships between markets as well as give a clearer insight to volatility.

The conclusion will not mention the mode decomposition techniques (EMD) or any relation between SV models and EMD.

Round 3

Reviewer 1 Report

Dear author, it is very much appreciated the effort made on this manuscript and the vast amount of work needed to arrive at this point. However, after our reviews, we find that unfortunately, the document is not round and sharp enough after all the discussion.

The main problem we see is that suggested changes in both revisions have not been properly answered or incorporated what results in a confusing outcome.

We suggest a careful revision and reorganization for future publication.

Author Response

I accept your decision.